# A convenient polyculture system that controls a shrimp viral disease with a high transmission rate

Muhua Wang [1,2,11], Yonggui Chen[1,2,11], Zhong Zhao[3,11], Shaoping Weng[1,2,4,5,11], Jinchuan Yang[4], Shangyun Liu[4], Chang Liu[4], Fenghua Yuan[1], Bin Ai[1], Haiqing Zhang[1], Mingyan Zhang[1], Lirong Lu[1], Kai Yuan[1], Zhaolong Yu[5], Bibo Mo[1], Xinjian Liu[6], Chunlei Gai[7], Yijun Li[8], Renjie Lu[9], Zhiwei Zhong[4], Luwei Zheng[1], Guocan Feng [3,12✉], Shengwen Calvin Li [10,12✉] & Jianguo He [1,2,4,5,12✉]

Developing ecological approaches for disease control is critical for future sustainable aquaculture development. White spot syndrome (WSS), caused by white spot syndrome virus (WSSV), is the most severe disease in cultured shrimp production. Culturing specific pathogen-free (SPF) broodstock is an effective and widely used strategy for controlling WSS. However, most small-scale farmers, who predominate shrimp aquaculture in developing countries, cannot cultivate SPF shrimp, as they do not have the required infrastructure and skills. Thus, these producers are more vulnerable to WSS outbreaks than industrial farms. Here we developed a shrimp polyculture system that prevents WSS outbreaks by introducing specific fish species. The system is easy to implement and requires no special biosecurity measures. The promotion of this system in China demonstrated that it allowed small-scale farmers to improve their livelihood through shrimp cultivation by controlling WSS outbreaks and increasing the production of ponds.

[1] State Key Laboratory for Biocontrol, School of Marine Sciences, Sun Yat-Sen University, Zhuhai 519000, China. [2] Southern Marine Science and Engineering Guangdong Laboratory (Zhuhai), Zhuhai 519000, China. [3] School of Mathematics, Sun Yat-Sen University, Guangzhou 510275, China. [4] School of Life Sciences, Sun Yat-Sen University, Guangzhou 510275, China. [5] Maoming Branch, Guangdong Laboratory for Lingnan Modern Agricultural Science and Technology, Maoming 525435, China. [6] Guangzhou Nansha District Yinong fishery cooperative, Guangzhou 511464, China. [7] Marine Science Research Institute of Shandong Province, Qingdao 266104, China. [8] Hainan Changjiang Nanjiang Biotechnology Co., Ltd., Changjiang 572700, China. [9] Aquatic Fine Breed & Fisheries Environmental Monitoring and Protection Center of Hebei Province, Shijiazhuang 050035, China. [10] University of California-Irvine School of Medicine, Children's Hospital of Orange County, Orange, CA 92868-3874, USA. [11] These authors contributed equally: Muhua Wang, Yonggui Chen, Zhong Zhao, Shaoping Weng. [12] These authors jointly supervised this work: Guocan Feng, Shengwen Calvin Li, Jianguo He. ✉email: mcsfgc@mail.sysu.edu.cn; shengwel@uci.edu; lsshjg@mail.sysu.edu.cn

Due to the decline of wild fishery resources worldwide, aquaculture plays a critical role in meeting the increasing demand for aquatic foods, which are a major source of animal protein[1,2]. In addition, aquaculture can improve the socioeconomic condition and livelihood of people in low-income countries by providing a highly nutritious food supply, employment, and income[3,4]. In Asian countries, shrimp are predominantly cultivated by small-scale farmers[5,6]. Furthermore, shrimp farming has been adopted as a strategy to promote economic growth and alleviate the poverty of farmers in these countries[7]. The increasing incidence of infectious diseases outbreaks is a major problem affecting the expansion of the shrimp aquaculture industry[3]. Therefore, developing convenient and ecological approaches for small-scale farmers to control the disease is critical for the future sustainable development of the shrimp aquaculture industry and poverty alleviation in developing countries.

White spot syndrome (WSS), which is caused by the WSS virus (WSSV), leads to catastrophic economic losses for the global shrimp aquaculture industry of over $1 billion annually, outweighing the losses due to other major crustacean diseases[8,9]. WSS pandemics primarily occur with the sequential transmission of WSSV from healthy shrimp that consume dead WSSV-infected shrimp to other healthy shrimp[10,11]. Because of the high efficiency and low negative environmental impact, culturing specific pathogen-free (SPF) shrimp is the most widely used strategy for controlling WSS outbreaks[12]. Disease prevention by using SPF shrimp is only likely to be successful if accompanied by stringent and sophisticated pathogen-exclusion management practices[13]. However, small-scale farmers, especially those from low-income countries, have limited access to or cannot afford SPF broodstock. Moreover, they do not have the infrastructure and technical skills to apply the required biosecure practices for culturing SPF shrimp[14,15]. Therefore, these limited-resource farms, which cultivate shrimp to improve livelihoods, are more vulnerable to WSS outbreaks than industrial farms. Most of these small farms have suffered from financial collapse due to production losses caused by WSS outbreaks[16].

Polyculture in aquaculture, which is cultivating more than one species in the same pond, might maximize yield and reduce wastes in effluent through better utilization of the available food in the system[17,18]. Therefore, polyculture has been considered as a promising strategy in the future sustainable shrimp aquaculture industry[18]. As general theory predicts that selective predation on infected individuals can reduce the prevalence of diseases in the prey population[19,20], polyculture might prevent WSSV outbreaks by restoring the spatiotemporal interaction of predators and prey. Here we developed a cost-effective and convenient shrimp polyculture system that effectively prevents outbreaks of WSS by introducing specific fish. The system is highly robust and has been demonstrated to successfully control WSS outbreaks in the cultivation of major cultivated marine shrimp species, including Pacific white shrimp (Litopenaeus vannamei), black tiger shrimp (Penaeus monodon), kuruma shrimp (Marsupenaeus japonica), and Chinese white shrimp (Fenneropenaeus chinensis). The implementation of this polyculture system does not require taking biosecurity measures. Furthermore, the system is capable of controlling WSS outbreaks even when there are WSSV carriers in shrimp postlarvae. Thus, small-scale farms can easily adopt this system to control WSS outbreaks without additional investment.

## Results

### The transmission dynamics of WSS in a shrimp population.
The effectiveness of selective predation in achieving disease prevention is determined by the interplay of several factors[21–23].

The prevalence of diseases in the prey population is positively correlated with the disease transmission rate but is negatively correlated with predation pressure and predator selectivity[24,25]. Thus, to develop a shrimp culture system that controls WSSV outbreaks through selective predation, we studied WSSV transmission dynamics in an L. vannamei population by determining the relationships among the bodyweight of one initial WSSV-infected shrimp, a number of deaths, and death time distribution. One piece of dead WSSV-infected shrimp infected a large number of healthy shrimp with the same bodyweight via ingestion. The number of infected shrimp in the groups exhibiting average body weights of 1.98, 6.13, and 7.95 g was 57.3, 64.7, and 71.3, respectively. This suggests that the transmission rate of WSSV is extremely high, and the basic reproduction number ($R_0$) of WSSV increases with the bodyweight of dead WSSV-infected shrimp (Supplementary Fig. 1 and Supplementary Table 1). Time to death was consistent across the three groups of body weights for WSSV-infected shrimp, with the majority of deaths occurring on the third to sixth day and the peak number of deaths occurring on the fourth and fifth days (Fig. 1a; Supplementary Fig. 2; Supplementary Tables 2–4). A mathematical model (Model 1) was developed to describe the transmission dynamics of WSS (Fig. 1a, b). In addition, the changes in live and dead shrimp numbers during WSSV transmission were determined by artificial infection experiments. The number of live shrimp began to decrease 2 days after WSSV infection and drastically decreased 4 days after WSSV infection (Fig. 1c and Supplementary Table 5). The dynamic changes in healthy, infected, and dead shrimp could be expressed by a mathematical model (Model 2) (Supplementary Fig. 3). Model 2 predicts that it is possible to cut off the transmission route of WSSV by removing infected and dead shrimp, but the time window for prevention is approximately 2 days (Supplementary Fig. 4).

### Polyculture system for controlling WSS in L. vannamei cultivation.
We first developed a polyculture system for L. vannamei, as it is the primary cultivated shrimp species. To identify the fish species for controlling WSSV transmission, we examined the feeding ability and selectivity of diverse fish species that ingest L. vannamei, including grass carp (Ctenopharyngodon idella), African sharptooth catfish (Clarias gariepinus) (hereafter referred to as catfish), and red drum (Sciaenops ocellatus). The daily dead shrimp ingestion rates of grass carp, catfish, and red drum were 8.26%, 4.99%, and 11.63%, respectively (Supplementary Tables 6–8). The daily healthy shrimp ingestion rates of grass carp, catfish, and red drum were 2.09%, 1.01%, and 6.04%, respectively, indicating the relatively high healthy shrimp ingestion rate of red drum (Supplementary Tables 9–11). In addition, grass carp and catfish have high feeding selectivity of dead shrimp over infected and healthy shrimp (Fig. 2a; Supplementary Fig. 5; Supplementary Tables 12 and 13). These characteristics suggest that grass carp and catfish have high feeding selectivity and ability, which can cut off the WSSV transmission route in which healthy shrimp ingest dead WSSV-infected shrimp.

If fish could not swallow intact WSSV-infected shrimp, healthy shrimp might ingest the remaining parts of the infected shrimp, which would result in the transmission of WSSV. Thus, we identified the suitable bodyweight of cocultured fish for WSS prevention. Experimental ponds were set up in which 600 healthy and 3 WSSV-infected shrimp were cultured with one grass carp of different body weights. After 13 days in culture, the ponds cocultured with one grass carp weighting 0.3, 0.5, 1, and 1.5 kg showed shrimp survival rates of 0, 0, 82.4%, and 79.4%, respectively (Fig. 2c; Supplementary Table 14). This finding indicates that the suitable bodyweight of grass carp for effective

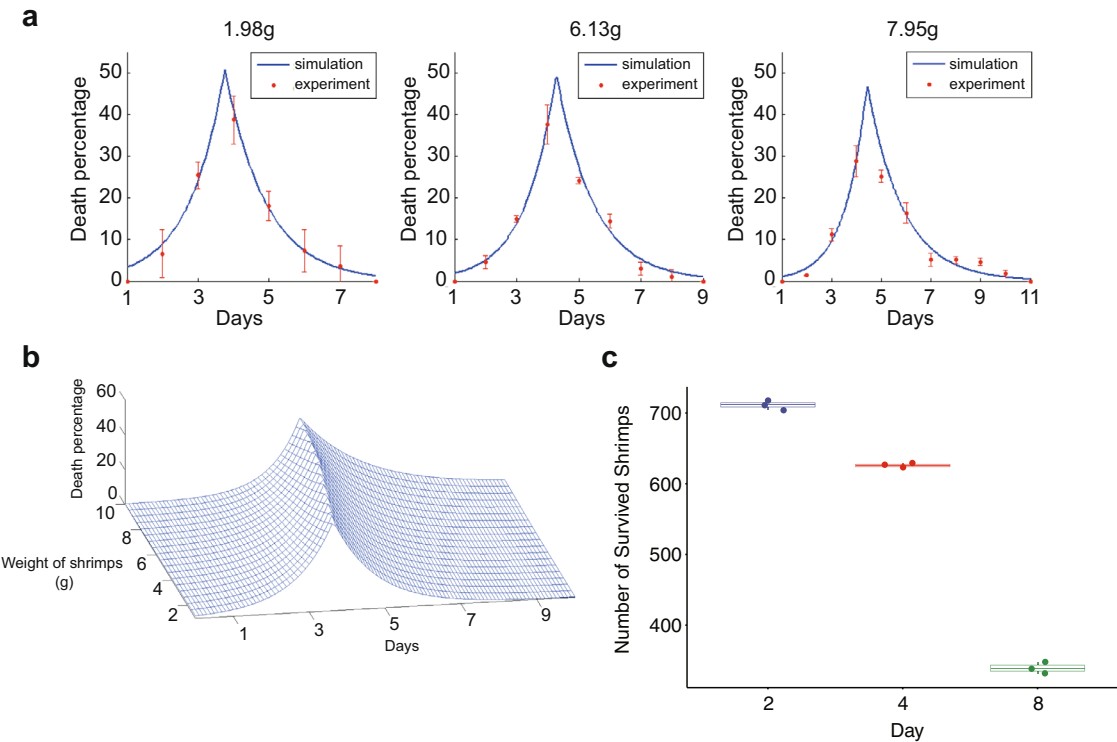

**Fig. 1 Transmission dynamics of WSSV. a** Daily death percentage of shrimp populations with initial WSSV-infected shrimp of different body weights. The red points with error bars are the results from experiments, and the solid blue lines are the results of model 1. **b** The illustration shows the death percentages of infected shrimp with different body weights each day. We estimate the parameters in model l and draw the above 3D surface to show the relationship between the death percentage concerning shrimp weights and time. For all shrimp weights, the death percentage rises at the beginning and then drops. The peak time for death was approximately the fourth day. **c** The relationship between the number of surviving shrimp and days after healthy shrimp were cocultured with WSSV-infected shrimp. Means and standard errors are shown ($n = 3$).

control of WSS is 1 kg. Furthermore, experimental ponds were set up in which 600 healthy and WSSV-carrying shrimp and 3 artificially WSSV-infected shrimp with the same body weights were cultured with one catfish of different body weights. After 14 days in culture, the ponds cocultured with one catfish weighting 0.25, 0.5, 0.75, and 1.5 kg showed shrimp survival rates of 19%, 40.83%, 48.67%, and 55.5%, respectively (Supplementary Fig. 6; Supplementary Table 15). Moreover, nearly all dead shrimp were removed by fish in the ponds cocultured with catfish of bodyweights greater than 0.5 kg. These results indicate that the suitable bodyweight of catfish for effective control of WSS is 0.5 kg. In addition, coculturing shrimp and fish can control WSS outbreaks even when there are WSSV carriers in shrimp postlarvae.

Next, we determined the capacity of grass carp for WSS prevention in shrimp populations of different shrimp body weights based on experimental results. In a 10-m² pond with 750 shrimp, one 1-kg grass carp could control 70 pieces of 2.5 g WSSV-infected shrimp, 50 pieces of 5.0 g WSSV-infected shrimp, or 30 pieces of 7.8 g WSSV-infected shrimp (Supplementary Table 16). This result suggests that the capacity of grass carp to control WSS is negatively correlated with the bodyweight of shrimp. Thus, releasing fishes in the early stages of shrimp production may improve their capacity to control WSS. In addition, the experimental result is consistent with the result of the mathematical simulation (Model 3) (Fig. 2c). Model 3 shows that the dynamics of the infected shrimp are related to the number of healthy shrimp being infected, the number of deaths of infected shrimp, and the number of infected shrimp eaten by fish. Therefore, polyculture should control WSS outbreaks in the cultivation of diverse cultivated shrimp species as long as the fish

can swallow the WSSV-infected and dead shrimp promptly and completely. In addition, the capacity of fish for preventing WSS outbreaks can be determined by Model 3, which is suitable for diverse species of fish.

Next, we translated the knowledge obtained from the experiments to an applied technology scale. The minimum stocking quantities of grass carp and catfish to effectively control WSS were determined to be 300 grass carp/ha and 600 catfish/ha by experiments (Fig. 2d; Supplementary Fig. 7; Supplementary Table 17 and 18). Accordingly, we developed two polyculture systems for preventing WSS outbreaks in *L. vannamei* cultivation by coculturing grass carp and catfish (Supplementary Note). The effectiveness of controlling WSSV transmission by coculturing shrimp with grass carp was tested in two experimental zones at Farm 1 in 2011 (Fig. 3a). In the 18 ponds (6.03 ha) of zone A, we cultured $9 \times 10^5$/ha shrimp for 20 days and then introduced 317–450/ha grass carp with an average body weight of 1 kg. We did not observe WSS outbreaks in 17 ponds and harvested $7,332 \pm 2,059$ kg/ha of shrimp in 110 days of culture (Fig. 3b; Supplementary Table 19). One pond was unsuccessful due to pathogenic bacterium (Vibrio) infection. Shrimp were cultivated without grass carp in 28 ponds (11.30 ha) in zone B. A total of 20 ponds in zone B had WSS outbreaks, resulting in an average yield of $1844 \pm 1034$ kg/ha. In 2012, we switched zones A and B, cultivating shrimp with grass carp in zone B but without fish in zone A. We did not observe WSS outbreaks in any of the ponds in zone B during 110 days of culture, while 12 of 18 ponds in zone A had WSS outbreaks.

To evaluate the effectiveness of controlling WSSV transmission by coculturing shrimp with catfish, two experimental zones were designed at Farm 2 in 2011 (Fig. 3c). In zone A, we cultivated

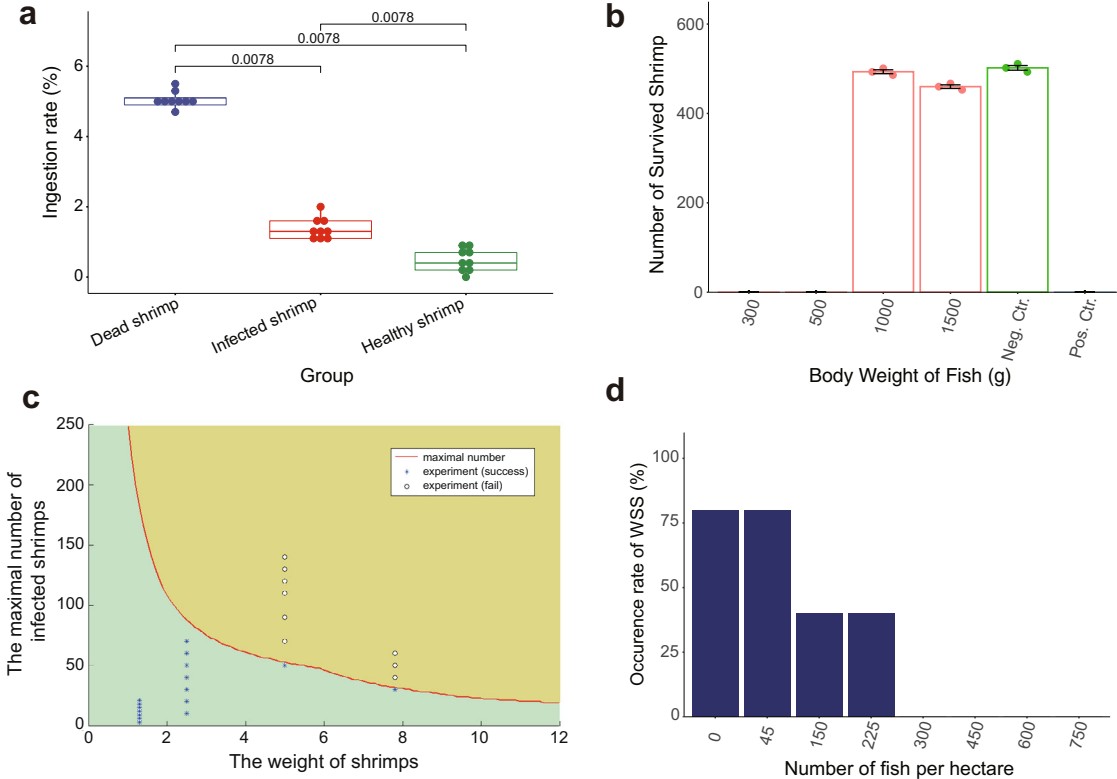

**Fig. 2 Specification of grass carp for the control of WSSV transmission. a** Feeding selectivity of grass carp on dead, infected (endopod and exopod removed), and healthy shrimp. The diseased shrimp infected with WSSV died within two days, which makes it hard to distinguish the initial dead shrimp from the ones that were died from diseased shrimp. The diseased shrimp had reduced activity, and the activity of shrimp was reduced after the endopods and exopods were removed. Thus, the shrimp with endopods and exopods removed were utilized to resemble WSSV-infected shrimp. *P*-values (permutation test, paired) were labeled (*n* = 9). Grass carp ingested significantly more dead shrimp than infected (endopod and exopod removed) and healthy shrimp. **b** The effect of the different body weights of grass carp on the control of WSS outbreaks. Means and standard errors are shown (*n* = 3). **c** Capacity of 1-kg grass carp to control WSS. Blue asterisks represent the number of infected shrimp successfully controlled by one 1-kg grass carp, while clear circles represent the number of infected shrimp that failed to be controlled by one 1-kg grass carp. The red line is the simulated highest value of one 1-kg fish that can control the number of infected shrimp with different body weights based on model 3. **d** The relationship of the number of cocultured grass carp and the occurrence rate of WSS. More than 300 grass carp of approximately 1.0 kg per hectare can completely control WSS outbreaks, but fewer than 225 grass carp cannot fully control WSS outbreaks.

$7.5 \times 10^5$/ha shrimp in 38 ponds (21.20 ha) for 10 days and then introduced 525–750/ha catfish with an average body weight of 0.5 kg. Shrimp were cultivated in 57 ponds (67.00 ha) without fish in zone B. In zone A, we did not observe WSS outbreaks in any of the ponds and harvested 8730 ± 1187 kg/ha of shrimp in 110 days of culture, while WSS outbreaks occurred in 53 ponds of zone B (Fig. 3e; Supplementary Table 20). In 2012, we split zone B into zones B1 and B2. Shrimp were cultivated with catfish in 38 ponds of zone A and 25 ponds (27.00 ha) of zone B1, while shrimp were cultivated without fish in 32 ponds (40.00 ha) of zone B2 (Fig. 3d). We did not observe WSS outbreaks in zone A or zone B1. However, WSS outbreaks were observed in 29 of 32 ponds in zone B2. To determine the effectiveness and usability of the shrimp polyculture system, we requested that Farm 1 cultivate shrimp with and without grass carp and/or catfish from 2013 to 2019. Releasing grass carp and/or catfish effectively controlled the WSS outbreak and substantially increased shrimp production (Fig. 3f; Supplementary Data 1).

**Polyculture system for controlling WSS in the cultivation of other species of shrimp.** We further developed polyculture systems for three widely cultivated shrimp species, *P. monodon*, *M. japonica*, and *F. chinensis*. Brown-marbled grouper (*Epinephelus fuscoguttatus*) was selected as the coculture fish in the polyculture

system of *P. monodon* (Supplementary Note). The effectiveness of controlling WSSV transmission by coculturing shrimp with brown-marbled grouper was tested at Farm 3 in 2013 and 2014 (Supplementary Fig. 8; Supplementary Table 21). We cultured $6 \times 10^5$/ha of non-SPF shrimp in 6 ponds (1.60 ha) for 30 days and then introduced 600–750/ha of brown-marbled grouper with an average body weight of 0.1 kg. We did not observe WSS outbreaks in any of the ponds in 150 days of culture and harvested 6395 ± 427 kg/ha and 6440 ± 447 kg/ha of shrimp in 2013 and 2014, respectively. However, WSS outbreaks occurred in all 3 ponds (0.80 ha) in which shrimp were cultivated without fish in these two years.

We selected branded gobies (*Chaeturichthys stigmatias*) as the coculture fish in the polyculture systems of *M. japonica* or *F. chinensis* (Supplementary Note). The effectiveness of controlling WSSV transmission in *M. japonica* cultivation by coculturing branded gobies was tested at Farm 4 in 2013 and 2014 (Supplementary Fig. 9; Supplementary Table 22). We cultured $1.5 \times 10^5$/ha of non-SPF shrimp in 10 ponds (13.40 ha) for 30 days and then introduced 750–900/ha of branded gobies with an average body weight of 0.05 kg. We did not observe WSS outbreaks in any of the ponds in 100 days of culture and harvested 1089 ± 50 kg/ha and 1121 ± 48 kg/ha of shrimp in 2013 and 2014, respectively. Shrimp were also cultivated without fish in 5 ponds (6.70 ha) as a control. In 2013, 4 out of 5 ponds

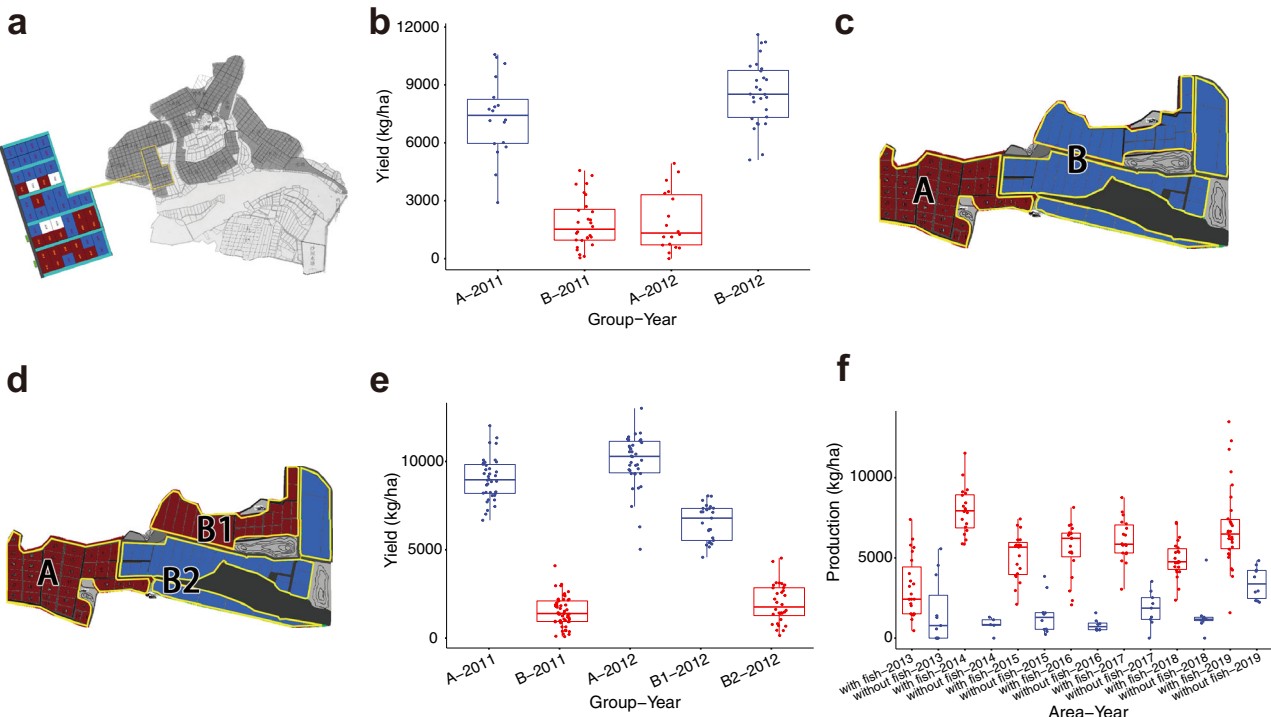

**Fig. 3 Control of WSS in *L. vannamei* production by fish. a** Design of the field study for the control of WSS using grass carp. The satellite map of the farm at Maoming, Guangdong Province, China, (Farm 1) is shown. The 46 experimental ponds were divided into zone A (red) and zone B (blue). In 2011, shrimp were cultured with grass carp in ponds in area A, while shrimp were cultured without fish in ponds in area B. In 2012, shrimp were cocultured with grass carp in area B but without fish in area A. **b** Total yield of shrimp production in ponds with (red) or without grass carp (blue) at Farm 1. **c** Design of the field study for the control of WSS using catfish in 2011. The satellite map of the farm in Qinzhou, Guangxi Province, China, (Farm 2) is shown. The 95 experimental ponds were divided into zone A (red) and zone B (blue). Shrimps were cultured with catfish in the ponds in area A, while shrimp were cultured without fish in the ponds in area B. **d** The design of the field study for the control of WSS using catfish in 2012. Shrimp continued to be cultured with catfish in the ponds in area A, while area B was divided into two groups: shrimp were cultured with catfish in the ponds in area B1, and shrimp were cultured without fish in the ponds in area B2. **e** Total yield of shrimp production in ponds with (red) or without catfish (blue) at Farm 2. **f** Total yield of shrimp production in ponds with (red) or without fish (blue) at Farm 1 from 2013 to 2019.

had WSS outbreaks. In 2014, WSS outbreaks were observed in all 5 ponds, resulting in an average yield of 407 ± 16 kg/ha.

**The polyculture system alleviates the poverty of small-scale farmers**. The promotion of the system in China demonstrated that it could alleviate the poverty of small-scale farmers. First, the polyculture system can control WSS outbreaks in the production of major cultivated marine shrimp species. Farmers can adapt the system to cultivate various shrimp species under different conditions. The system has been adopted by farmers in ten provinces of China (Liaoning, Hebei, Tianjin, Shandong, Jiangsu, Zhejiang, Fujian, Guangdong, Guangxi, Hainan). Second, the implementation does not require applying any biosecurity measure, and the fish used in the system are common aquaculture species or easy to find in coastal areas. Thus, small-scale farms can easily adopt the system in earthen ponds to prevent WSS outbreaks without additional investment. During the promotion in 2015, six farmers at a farmers' association in Nansha, China, decided to adopt the polyculture system in their earthen ponds without infrastructure renovation (Supplementary Fig. 10). Other than introducing fish in the pond, these farmers cultivated non-SPF shrimp as they usually do. These farmers harvested 3732 ± 510 kg of *P. monodon* as well as 6267 ± 236 kg of grass carp, while other farmers suffered from production losses caused by WSS outbreaks (Fig. 4a; Supplementary Table 23). Thus, all the farmers in the association started to use the system in 2016. WSS outbreaks have not been reported in the association since then. Third, controlling WSS outbreaks allows farmers to intensify shrimp production,

which leads to high productivity per unit area. For instance, farmers at a farmers' association in Tanghai, China, cultivated 1500/ha of *F. chinensis* in ponds of 5 ha in 2014, as increasing stocking quantity led to the outbreak of WSS (Supplementary Fig. 11). After the promotion of the polyculture system in 2015, the farmers cultivated 8000/ha of shrimp in the ponds, which substantially increased the yield of shrimp from 175 ± 19 kg/ha to 1159 ± 135 kg/ha (Fig. 4b; Supplementary Table 24).

## Discussion

The aquaculture industry is predicted to play a critical role in fulfilling the fast-growing demand for animal protein in 2050[26]. It is important to recognize that small-scale farms predominate aquaculture in developing countries[27]. The implementation of an ecological approach to disease control in small-scale aquaculture is critical for the sustainable development of the aquaculture industry[28]. Cultivating SPF broodstock is an efficient and sustainable approach to control viral diseases in crustacean aquaculture. However, most small-scale farmers in developing countries cannot afford expensive SPF broodstock or do not have the infrastructure and skills to perform the stringent and sophisticated biosecurity practices required to cultivate SPF stock. Therefore, in addition to being efficient and sustainable, the disease-control strategies developed for small-scale farms have to be cost-effective and easy to implement.

A positive spatiotemporal interaction among species exists in nature, which helps sustain food webs and control certain epidemic diseases[29–32]. Intensive monoculture in aquaculture leads

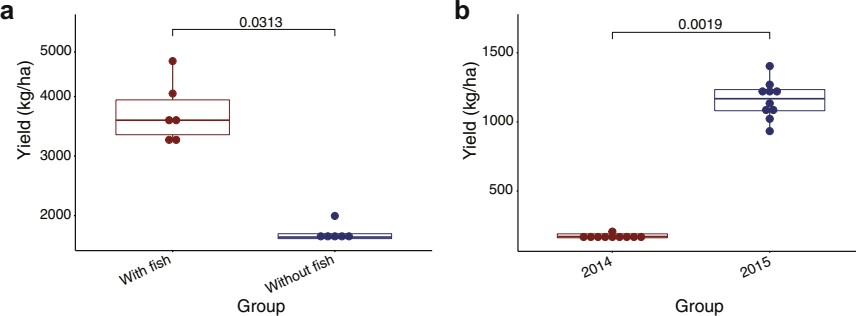

**Fig. 4 Shrimp polyculture systems alleviate the poverty of small-scale farmers. a** Total yield of *P. monodon* production in ponds with (red) or without grass carp (blue) at the farmers' association in Nansha, China. *P*-values (permutation test, paired) were labeled (*n* = 6). **b** Total yield of *F. chinensis* production in ponds at the farmers' association in Tanghai, China, in 2014 (red) and 2015 (blue). Shrimp were cultivated without fish in 2014 and with fish in 2015. *P*-values (permutation test, paired) were labeled (*n* = 10).

to high productivity per unit area but also eliminates the interactions among species that occur in ecosystems[33–35]. Once some individuals in the pond carry pathogens, severe disease outbreaks might occur quickly due to the high density of hosts and lack of species interactions[36]. Theory predicts that selective predation on infected individuals can control disease in the prey population but not for diseases with high transmission rates[19,21]. WSS is a severe disease with a high transmission rate. WSSV replicates rapidly and can result in cumulative mortality of up to 100% within 7–10 days in farmed shrimp[37,38]. By elucidating the transmission dynamics of WSSV, we found that there is a short time window for controlling the virus through selective predation. As the prevalence of diseases in the prey population is positively correlated with the disease transmission rate but negatively correlated with predation pressure and predator selectivity[24,25], we identified the species of cocultured fish by determining the feeding ability and selectivity of fish for healthy, WSSV-infected, and dead shrimp. Accordingly, we developed a convenient shrimp polyculture system that successfully controls WSS outbreaks by simply introducing specific fish species. Our results highlight the significance of determining the transmission dynamics of diseases in developing disease-control strategies through selective predation. In addition, this report demonstrates that polyculture, a traditional aquaculture practice, has the potential to control disease outbreaks by restoring the interactions of predators and prey.

The polyculture system plays a critical role in the future sustainable development of the shrimp aquaculture industry by providing a cost-effective and convenient approach to controlling WSS outbreaks for small-scale farmers, which could facilitate poverty alleviation in developing countries. Cultivating SPF broodstock is an effective and sustainable approach to preventing WSS outbreaks[12]. In addition, the sanitary status allows SPF stocks to be cultivated in a super-intensive manner, which makes the cultivation of SPF stocks highly profitable. However, most small-scale farms, which predominate shrimp aquaculture in Asia, cannot afford and do not have the infrastructure and ability to cultivate SPF stocks, which makes them more vulnerable to WSS outbreaks than industrial farms. Furthermore, there is no SPF broodstock available for several cultivated shrimp species, including *M. japonica* and *F. chinensis*[12]. This hampers small-scale farmers from improving their livelihood by cultivating indigenous shrimp species. Although it cannot prevent outbreaks of WSS in super-intensive shrimp cultivation at an industrial farm, the polyculture system described in this report is suitable for small-scale farms to alleviate poverty through shrimp aquaculture. First, the system is highly robust and can prevent outbreaks of WSS in the cultivation of major marine cultured shrimp species. Second, this system is cost-effective and easy to

implement, and it can control WSS outbreaks even when there are WSSV carriers in postlarvae. Finally, the production of ponds can be increased by intensifying shrimp production and harvesting shrimp and fish simultaneously. In sum, this system provides an example of sustainable ecological production in aquaculture by controlling WSS outbreaks, alleviating the poverty of small-scale farmers, and reducing the environmental impact of shrimp farming.

The polyculture system described in this report is highly robust, which can prevent WSS outbreaks of diverse cultivated species of shrimps by introducing different fish species. To develop the polyculture system, transmission dynamics of WSSV and dynamics of the WSSV-infected shrimp were determined through experiments and mathematical modeling. Model 2 showed that there is a short time window (2 days) for preventing WSS outbreaks. As the time window is too short, it is necessary to coculture shrimp with fish, which allows the fish to remove the moribund shrimp promptly. In addition, Model 3 demonstrated that a polyculture system can effectively control WSS outbreaks in shrimp cultivation as long as the fish can swallow the WSS-infected and dead shrimp promptly and completely. Thus, new polyculture systems can be developed based on the results of experiments and our mathematical models.

Farmers usually cultivate shrimps at low salinity in the estuary areas of China, where produces more than 50% of shrimps in China. Therefore, two freshwater fish species, grass carp and catfish, were used in our polyculture systems. We also developed polyculture systems using marine fish species (brown-marbled grouper and branded goby). Farmers can select a polyculture system that is suitable for their cultivation conditions. In addition, new polyculture systems can be developed to adapt to local cultivation conditions by changing the co-cultured fish species. Local aquaculture or native fish species are recommended to be used in the polyculture systems, as they are well-adapted to the local environment. The use of exotic fish species in the polyculture system should be highly cautious and certified by local administration agencies, as it might cause adverse environmental impacts. It is reported that shrimps can be infected by a few fish diseases[39]. Thus, the co-culture fish species should not carry pathogens that can infect shrimps. Otherwise, the fishes should be screened for these pathogens before introducing into the ponds.

## Methods
**Mathematical model 1—the relationship among the bodyweight of the initial WSSV-infected shrimp, number of deaths, and death time distribution.** The experimental data show the time course of death for the infected shrimp satisfies the Laplacian distribution (Supplementary Tables 2–4). The relationship of the bodyweight of the initial infected shrimp number of deaths and death time

distribution could be expressed by a mathematical model and the establishment of the mathematical model as shown below.

Suppose that one dead shrimp could infect $n$ healthy shrimp at the same day. These $n$ infected shrimp do not die simultaneously but on different days (time course). The value of $n$ is related to the weight of the dead shrimps—larger dead shrimp can infect more healthy shrimps of the same body weight. Our experimental results (Supplementary Tables 2–4) show the death time course for these $n$ infected shrimp satisfies the Laplacian distribution, as follows:

$$p(t) = \begin{cases} bexp\left(-\frac{|t-a|}{c_1}\right), t \le a \\ bexp\left(-\frac{|t-a|}{c_2}\right), t > a \end{cases} \quad (1)$$

where $a$ is the peak time of number of dead shrimps, $b$ is the maximal death percentage, $c_1$ is related to the mortality increases of the infected shrimps, $c_2$ is related to the mortality decreases of the infected shrimp, $p(t)$ is the percentage of infected shrimp that die at time $t$. The open bracket "{" in formula (1) means the function is represented by two parallel expressions as described previously.

Based on the Supplementary Tables 2–4, we can determine the value of $a$, $b$, $c_1$, and $c_2$ by the least square estimation method. As different weight corresponds to different distribution of death time, we can compute the relationship of weight of death shrimps with corresponding $a$, $b$, $c_1$, and $c_2$ (Supplementary Table 25).

We found the relationship of $w$ with $a$, or $b$ or $c_1$ or $c_2$ is quadratic (Eq. 2), with the data in Supplementary Table 25, we have

$$\begin{cases} a = -0.0918w^2 + 0.8772w + 3.3449 \\ b = 0.0029w^2 - 0.0369w + 0.5849 \\ c_1 = -0.0186w^2 + 0.1739w + 0.7063 \\ c_2 = 0.0002w^2 + 0.0108w + 1.0827 \end{cases} \quad (2)$$

Using Model 1, we can predict the effects of different body weights of dead WSSV-infected shrimp through the ingestion pathway of WSSV-infected dead shrimp on the WSSV transmission rate.

**Mathematical model 2—the dynamic changes of healthy, infected, and dead shrimp during WSSV transmission.** We derived and established Model 2 to simulate the WSS transmission dynamics in cultured shrimp. Using Model 2, we predicted the dynamic changes of three states (healthy, infected, and dead shrimps) in cultured shrimp as influenced by the WSS epidemic with the following:

Now we can develop a model for the spread and break out of WSS. For any given weight $w$ of shrimps, let $s_h(t)$, $s_i(t)$, and $s_d(t)$ be the number of healthy shrimp, infected shrimp and dead shrimp respectively at time $t$. Let $I(t)$, $d(t)$ be the number of daily infected shrimp, daily dead shrimp, respectively, at time $t$.

According to infection process, the decrement of healthy shrimp is caused by their infection, therefore we have $\frac{ds_h}{dt} = -I(t)$. The quantity change of infected shrimp includes the infection of healthy shrimp and the death of infected shrimp, we have $\frac{ds_i}{dt} = I(t) - d(t)$. The increment of dead shrimp is caused by the death of the infected shrimp; thus we have $\frac{ds_d}{dt} = d(t)$. We obtain the following system of ordinary differential equations:

$$\begin{cases} \frac{ds_h}{dt} = -I(t) \\ \frac{ds_i}{dt} = I(t) - d(t) \\ \frac{ds_d}{dt} = d(t) \end{cases} \quad (3)$$

where $s_h(0) = s_{h_0}$, $s_i(0) = s_{i_0}$, $s_d(0) = s_{d_0}$ are as the initial value, at $t = 0$.

In the above system of ordinary differential equations, quantity $I(t)$ can be expressed as follows

$$I(t) = min\left\{ns_d(t), s_h(t) - \alpha s_{h_0}\right\} \quad (4)$$

$d(t)$ can be expressed as

$$d(t) = \int_0^T min\left\{ns_d(t-\tau), s_h(t-\tau) - \alpha s_{h_0}\right\} p(\tau) d\tau \quad (5)$$

where $n$ is the number of healthy shrimp infected by one dead shrimp on the first day. $p(\tau)$ is the death percentage of the $n$ infected shrimp on the $\tau$ days, $T$ is the longest survival time of infected shrimp.

Now we explain how to set up the formulas $I(t)$ and $d(t)$. In the expression of $I(t)$, $ns_d(t)$ is the number of daily infected shrimp at time $t$. But as the number of healthy shrimp decreases, there may not be as many as $ns_d(t)$ healthy shrimp to be infected, $I(t)$ is the minimum of $ns_d(t)$ and $s_h(t) - \alpha s_{h_0}$, where $\alpha (0 < \alpha < 1)$ represents the percentage of healthy shrimp that may have resistance to viruses, $d(t)$ is the number of shrimps infected from 0 to $t$ die at time $t$. We use this integral to express the number of shrimp die at time $t$.

To evaluate the performance of the model 2, we compare the simulated scenario and the biological experimental settings. Our experiments show the quantity change of dead shrimps and live shrimps with respect to time, which is consistent with the result of simulation (Supplementary Fig. 4).

**Mathematical model 3—use fish to control WSS.** We established Model 3 for the prevention and control of WSS using fish. In Model 3, two parameters need to be determined before this model can be applied for evaluating the fish's capability of WSS prevention and control. The two parameters are, (1) fish-feeding quantity of dead shrimp, and (2) fish-feeding ratio of dead shrimp over healthy shrimp. We obtained 1 kg grass carp's feeding quantity of different body weights of shrimp and the feeding selectivity through experiments. The mathematical reasoning of Model 3 is as follows:

To block the transmission of WSS, we apply fish to eat dead shrimp and infected shrimp. Let $e_h(t)$, $e_i(t)$, and $e_d(t)$, respectively be the number of healthy shrimp, infected shrimp and dead shrimp eaten by fish daily at time $t$, $f(t)$ is the number of fish.

The decrement of healthy shrimp is related to the number of infected healthy shrimp and the number of shrimp eaten by fish, as expressed in $\frac{ds_h}{dt} = -I(t) - e_h(t)$. Similarly, the dynamics of the infected shrimp is related to the number of infected healthy shrimp, the death number of infected shrimp, and the number of infected shrimp eaten by fish, as expressed in $\frac{ds_i}{dt} = I(t) - d(t) - e_i(t)$. The dynamics of dead shrimp is related to the death number of infected shrimp, and eaten by fish, as expressed in $\frac{ds_d}{dt} = d(t) - e_d(t)$. Combining the above formulae, we can write the model as follows:

$$\begin{cases} \frac{ds_h}{dt} = -I(t) - e_h(t) \\ \frac{ds_i}{dt} = I(t) - d(t) - e_i(t) \\ \frac{ds_d}{dt} = d(t) - e_d(t) \end{cases} \quad (6)$$

where $s_h(0) = s_{h_0}$, $s_i(0) = s_{i_0}$, $s_d(0) = s_{d_0}$ are as the initial value at $t = 0$. In the above model, $I(t)$, $d(t)$, $e_h(t)$, $e_i(t)$, and $e_d(t)$ are respectively given as follows:

$$\begin{cases} I(t) = min\left\{ns_d(t), s_h(t) - \alpha s_{h_0}\right\} \\ d(t) = \int_0^t min\left\{ns_d(t-\tau), s_h(t-\tau) - \alpha s_{h_0}\right\} p(\tau)exp\left\{\int_{t-\tau}^t lnr(u)du\right\}d\tau \\ e_d(t) = min\left\{f(t) \cdot m \cdot \beta, s_d(t) + d(t)\right\} \\ e_i(t) = min\left\{\left(f(t) \cdot m - e_d(t)\right)\frac{s_i(t)+I(t)-d(t)}{s_i(t)+s_h(t)-d(t)}, s_i(t) + I(t) - d(t)\right\} \\ e_h(t) = min\left\{f(t) \cdot m - e_d(t) - e_i(t), s_h(t) - I(t)\right\} \\ r(t) = 1 - \frac{e_i(t)}{s_i(t)+I(t)-d(t)} \end{cases} \quad (7)$$

where, $I(t)$ is the same as in Eq. (4); for $d(t)$, different from Eq. (5) is that we add an exponential item $exp\left\{\int_{t-\tau}^t lnr(u)du\right\}$ to account for the infected shrimp that may be eaten by fish during the past $t$ days. As for $e_d(t)$ shown in Eq. (6), $m$ is for that each fish eats $m$ shrimps while $\beta$ accounts for a percentage of dead shrimp in $m$ shrimp. In $e_i(t)$, we introduce $\frac{s_i(t)+I(t)-d(t)}{s_i(t)+s_h(t)-d(t)}$ for the percentage of infected shrimp in live shrimp. $e_h(t)$ accounts for the number of healthy shrimp eaten by fish. $r(t)$ represents the percentage of infected shrimp not being eaten by fish. We performed the effects of 1 kg grass carps on shrimp with four different body weights. The simulated data agreed with the experimental results (Fig. 2c).

**The relationship among the bodyweight of one initial WSSV-infected shrimp, number of deaths, and death time distribution.** Three groups of 430 shrimp with a bodyweight of 1.98 ± 0.03, 6.13 ± 0.16, and 7.95 ± 0.13 g, respectively, were used. In each group, 30 shrimp were randomly selected and subjected to a two-step WSSV PCR assay. All the tested shrimp showed negative in the assay. The remaining 400 shrimps were divided equally and introduced to three experimental and one control ponds. All 12 aquariums (220 cm × 60 cm × 80 cm) were set up with a water volume of 0.5 m³ and a salinity of 8‰. Shrimp were quarantined for seven days before the experiment started. One piece of dead WSSV-infected shrimp was then introduced to each of the experimental aquariums. In addition, one piece of frozen dead shrimp (WSSV-free) was introduced to the control aquarium. Shrimp were fed once a day with artificial feed that is 2% of their body weight. Shrimp feces were timely removed, and 50% of the water in the aquarium was exchanged every day. To prevent healthy shrimps from eating the moribund shrimp but not the initial dead WSSV-infected shrimp, shrimp were observed every 10 min to identify and remove moribund shrimp from the second day of the experiment. Moribund shrimp were identified as the ones having pleopod activity, but no response to glass rod agitation. The experiment was continued until three days after the appearance of the last moribund shrimp in each aquarium. Five pieces each of moribund and survived shrimps in each aquarium were subjected to a one-step WSSV PCR assay. All moribund shrimps showed WSSV-positive, while survived shrimps showed WSSV-negative. A mathematical model (Model 1) describing the relationship among the bodyweight of one initial WSSV-infected shrimp, number of deaths, and death time distribution was established based on the experimental results.

**The dynamic changes of live, infected, and dead shrimps during WSSV transmission.** To determine the changes in numbers of live and dead shrimp during WSSV transmission, 9 cement ponds (315 cm × 315 cm × 120 cm) were set

up with a water volume of 5 m³ and salinity of 8‰. Regarding the stocking quantity of $7.5 \times 10^5$/ha in shrimp farming production, 750 healthy shrimp with an average body weight of 7.9 g were cultured in each of the nine ponds.

To prepare the WSSV acute-infected shrimp, healthy shrimp were starved for 3 days, and then fed with parts of dead WSSV-infected shrimp that are 20% of their body weights twice a day. Five shrimp were randomly selected and subjected to a one-step WSSV PCR assay. If the tested shrimp showed WSSV positive in the assay. The rest of the shrimp in the aquarium was used as the WSSV acute-infected shrimp in the following experiments.

Healthy shrimp were quarantined for seven days before the experiment started. Thirty WSSV acute-infected shrimp were then introduced in each pond. Shrimps were fed once a day with artificial feed that is 2% of their body weight. The numbers of survived shrimp were counted in three ponds on the 2nd, 4th, 8th day after WSSV infection, respectively. Five dead shrimps in each pond were subjected to a one-step WSSV PCR assay, showing WSSV-positive. Based on model 1, we established a mathematical model (Model 2) to describe the dynamic changes of healthy, infected, and dead shrimps during WSSV transmission.

**The dead shrimp ingestion rate of fish.** To determine the dead shrimp ingestion rate of grass carp (*Ctenopharyngodon idellus*). Three cement ponds (315 cm × 315 cm × 120 cm) were set up with a water volume of 5 m³ and a salinity of 5‰. Three grass carps with an average body weight of 0.5 kg, 1 kg, and 1.5 kg were released in each of the three ponds, respectively. The fish were raised for four days and then fed with dead shrimps with an average weight of 5.3 g. In addition, to determine the dead shrimp ingestion rate of African sharptooth catfish (*Clarias gariepinus*). Four cement ponds (315 cm × 315 cm × 120 cm) were set up with a water volume of 5 m³ and salinity of 3‰. One African sharptooth catfish with bodyweight of 0.262, 0.496, 0.731, and 1.502 kg was released in each of the four ponds, respectively. The fish were raised for four days and then fed with dead shrimps with an average body weight of 6.2 g. Finally, to determine the dead shrimp ingestion rate of red drum (*Sciaenops ocellatus*). Three cement ponds (315 cm × 315 cm × 120 cm) were set up with water volume of 5 m³ and a salinity of 5‰. One red drum with a bodyweight of 0.590, 0.654, and 0.732 kg was released in each of the three ponds, respectively. The fish were raised for four days and then fed with dead shrimps with an average body weight of 3.9 g.

During the five days of the experiment, dead shrimp that were not ingested by fish were exchanged with new dead shrimps every day. Additionally, the total body weight of dead shrimp ingested by fishes was calculated by subtracting the total body weight of dead shrimp that remained in the pond from the total body weight of dead shrimp put in the pond. The shrimp ingestion rate of fish is quantified by the daily ingestion rate (total body weight of ingested shrimps per day/total body weight of fishes). The daily ingestion rate of fish was calculated for 5 days.

**The healthy shrimp ingestion rate of fish.** To determine the healthy shrimp ingestion rate of grass carp, three experimental ponds and one control pond (315 cm × 315 cm × 120 cm) were set up with a water volume of 5 m³ and salinity of 5‰. In total, 750 healthy shrimp with an average body weight of 5.3 g were cultured in each pond. One grass carp weighting 0.956, 1.013, and 1.050 kg was released in each of the experiment ponds, respectively. No fish was released in the control pond. Every two days, 50% of the water in each pond was changed. Live shrimp that remained in each pond were counted and weighted after 10 days of the experiment.

To determine the healthy shrimp ingestion rate of African sharptooth catfish, one experimental pond and one control pond (315 cm × 315 cm × 120 cm) were set up with a water volume of 5 m³ and salinity of 3‰. In total, 750 healthy shrimp with an average body weight of 2.2 g were cultured in each pond. One African sharptooth fish weighting 1.050 kg was released in the experiment pond. No fish was released in the control pond. Every 2 days, 50% of the water in each pond was changed. Live shrimp that remained in each pond were counted and weighted after 10 days of the experiment.

To determine the healthy shrimp ingestion rate of red drum, three experimental ponds and one control pond (315 cm × 315 cm × 120 cm) were set up with a water volume of 5 m³ and salinity of 5‰. In total, 750 healthy shrimp with an average body weight of 2.7 g were introduced in each pond. One red drum weighting 0.519, 0.554, and 0.595 kg was released in each of the experiment ponds, respectively. No fish was released in the control pond. Every two days, 50% of the water in each pond was changed. Live shrimp that remained in each pond were counted and weighted after 10 days of the experiment.

**The feeding selectivity of fish on dead, infected, and healthy shrimps.** To determine the feeding selectivity of grass carp on dead, infected, and healthy shrimp, one aquarium (220 cm × 60 cm × 80 cm) was set up with a water volume of 0.5 m³ and a salinity of 5‰. Grass carp weighting 1.58 kg was cultured in the aquarium for four days before the experiment started. The diseased shrimp infected with WSSV died within two days, which makes it hard to distinguish the initial dead shrimp from the ones that were died from diseased shrimp. The diseased shrimp had reduced activity, and the activity of shrimp was reduced after the endopods and exopods were removed. Thus, shrimp with endopods and exopods removed were utilized to resemble WSSV-infected shrimp. Thirty pieces each of

dead, WSSV-infected (endopods and exopods removed), and healthy shrimps were introduced in the aquarium. The mean weight of shrimp used in the experiment is 3.5 g.

To determine the feeding selectivity of African sharptooth catfish on dead, infected, and healthy shrimps, one aquarium (220 cm × 60 cm × 80 cm) was set up with a water volume of 0.5 m³ and salinity of 3‰. African sharptooth catfish with body weight of 1.03 kg was cultured in the aquarium for four days before the experiment started. Thirty pieces each of dead, WSSV-infected (endopods and exopods removed), and healthy shrimps were introduced in the aquarium. The mean weight of shrimps used in the experiment is 8.4 g.

During the 9 days of the experiment, the dead, infected (endopods and exopods removed), and healthy shrimp that remained in the aquarium were counted and weighed every day. New shrimps were added to ensure there are 30 pieces each of dead, infected (endopods and exopods removed), and healthy shrimp in the aquarium. The daily total body weight of shrimp that were ingested by fish in each pond was calculated by subtracting the total body weight of shrimp that remained in the pond from the total weight of shrimp put in the pond. The shrimp ingestion rate of fish is quantified by the daily ingestion rate (total body weight of ingested shrimp per day/bodyweight of fish).

**The suitable bodyweight of grass carp for controlling WSS.** To determine the suitable bodyweight of grass carp for controlling WSS, four experimental groups and two control groups were set up. Each group consisted of three ponds (315 cm × 315 cm × 120 cm). In total, 600 healthy and 3 WSSV-infected shrimp with an average body weight of 5 g were cultured in each pond of experimental groups. One grass carp with a bodyweight of 0.3, 0.5, 1.0, 1.5 kg was released in the ponds of each experimental group, respectively. In the positive control group, 600 healthy and 3 WSSV-infected shrimp with an average body weight of 5.0 g were cultured in each of the three ponds without introducing grass carp. In the negative control group, 600 healthy shrimp with an average body weight of 5.0 g were cocultured with one grass carp weighting 1.0 kg in each of the three ponds. The numbers of live shrimp were counted after ten days of the experiment. If there were dead shrimp in the ponds, they were subjected to a one-step WSSV PCR assay. All dead shrimps showed positive for WSSV infection.

**The suitable bodyweight of African sharptooth catfish for controlling WSS.** To determine the suitable bodyweight of African sharptooth catfish for controlling WSS, four experimental groups and two control groups were set up. Each group consisted of three ponds (315 cm × 315 cm × 120 cm). In total, 600 healthy and WSSV carrying shrimp and 3 WSSV-infected shrimp with an average body weight of 1.5 g were cultured in each pond of experimental groups. The WSSV carrying shrimp were determined as the ones that showed positive in a two-step WSSV assay. One African sharptooth catfish with a bodyweight of 0.25, 0.5, 0.75, 1.5 kg was released in the ponds of each experimental group, respectively. In the positive control group, 600 healthy and 3 WSSV-infected shrimp with an average body weight of 1.5 g were cultured in each of the three ponds without introducing African sharptooth catfish. In the negative control group, 600 healthy shrimps with an average body weight of 1.5 g were cocultured with one African sharptooth catfish weighting 1.0 kg in each pond. The numbers of live shrimp were counted after ten days of the experiment. If there were dead shrimps in the ponds, they were subjected to a one-step WSSV PCR assay. All dead shrimps showed positive for WSSV infection.

**The capacity of grass carp for controlling WSS.** To determine the capacity of grass carp for controlling WSS, the number of WSSV-infected shrimp that could be ingested by one grass carp weighting 1 kg was evaluated. Four groups of shrimp with different body weights (1.3 ± 0.1, 2.5 ± 0.2, 5.0 ± 0.3, 7.8 ± 0.5 g) were cocultured with 1-kg grass carp in the ponds.

In 1.3 ± 0.1 g group, 750 healthy shrimp were cultured in each of the nine cement ponds (315 cm × 315 cm × 120 cm). Healthy shrimps were cultured with 3, 6, 9, 12, 15, 18, and 21 pieces of WSSV-infected shrimp in each of the seven experimental ponds, respectively. One grass carp weighting 1 kg was released in each of the seven ponds. Healthy shrimp were cultured with 3 WSSV-infected shrimps in one pond as a positive control. Additionally, healthy shrimps were cultured without WSSV-infected shrimp nor grass carp in one pond as a negative control. In 2.5 ± 0.2 g group, 750 healthy shrimp were cultured with 10, 20, 30, 40, 50, 60, and 70 pieces of WSSV-infected shrimp in each of the seven experimental ponds, respectively. One grass carp weighting 1 kg was released in each of the seven ponds. Healthy shrimp were cultured with 10 WSSV-infected shrimp in one pond as a positive control. Additionally, healthy shrimps were cultured without WSSV-infected shrimp nor grass carp in one pond as a negative control. In 5.0 ± 0.3 g group, 750 healthy shrimp were cultivated with 50, 70, 90, 110, 120, 130, and 140 pieces of WSSV-infected shrimp in each of the seven experimental ponds, respectively. One grass carp weighting 1 kg was released in each of the seven ponds. Healthy shrimp were cultured with 50 WSSV-infected shrimps in one pond as a positive control. Additionally, healthy shrimps were cultured without WSSV-infected shrimps nor grass carp in one pond as a negative control. In 7.8 ± 0.5 g group, 750 healthy shrimp were cultured with 30, 40, 50, or 60 pieces of WSSV-infected shrimps in four experimental ponds, respectively. One grass carp

weighting 1 kg was released in each of the four ponds. Healthy shrimp were cultured with 30 WSSV-infected shrimps in one pond as a positive control. In addition, healthy shrimps were cultured without WSSV-infected shrimps nor grass carp in one pond as a negative control.

In all the ponds, shrimp were fed with artificial feed that is 2% of their body weight. And 50% of the water was changed every day. The numbers of the remaining live shrimp were counted after 15 days of the experiment. A mathematical model (Model 3) was established based on the relationship of healthy shrimp, infected shrimp, dead shrimp, and fish.

**Determine the numbers of grass carp and African sharptooth catfish required for controlling WSS in *L. vanmamei* cultivation**. The number of grass carp required for controlling WSS in shrimp production was determined in Pinggang Aquaculture Base, Yangjiang, China in 2010. Forty ponds (0.34 ± 0.04 ha/pond) were divided into eight groups; each group consisted of 5 ponds. We cultured 675,000/ha of shrimp in the ponds. Shrimp were cultured for 20 days before 45, 150, 225, 300, 450, 600, 750/ha of grass carp with an average body weight of 1.0 kg were released in the ponds of group 2 to group 8. Shrimp were cultured without fish in the ponds of group 1. These 40 ponds were managed by using the same farming method. If the WSS outbreak occurred, shrimps were harvested immediately; if not, shrimps were harvested after 110 days of cultivation.

The number of African sharptooth catfish required for controlling WSS in shrimp production was determined in Pinggang Aquaculture Base, Yangjiang, China in 2010. Thirty-five ponds (0.37 ± 0.06 ha/pond) were divided into seven groups; each group consisted of 5 ponds. We cultured 675,000/ha of shrimp in the ponds. Shrimp were cultured for 10 days before 150, 300, 450, 600, 750, 900/ha of African sharptooth catfish with an average body weight of 1.0 kg were released in the ponds of group 2 to group 7. Shrimp were cultured without fish in the ponds of group 1. These 35 ponds were managed by using the same farming method. If the WSS outbreak occurred, shrimps were harvested immediately; if not, shrimps were harvested after 110 days of cultivation.

**Validation of coculturing shrimp and grass carp for controlling WSS in *L. vanmamei* farming**. In 2011, the polyculture system of coculturing *L. vanmamei* and grass carps was validated at a farm in Maoming, Guangdong Province, China (Farm 1). Forty-six farm ponds (17.33 ha) were divided into zone A and zone B. Zone A consisted of 18 ponds with a total area of 6.03 ha, and zone B consisted of 28 ponds with a total area of 11.30 ha. The stocking quantity of shrimp in the ponds of zone A is 900,000/ha. Shrimp were cultured in the ponds for 20 days before releasing grass carps with an average body weight of 1.0 kg. The stocking quantity of fish is 317–450/ha. Shrimp were cultured without fish in the ponds of zone B, and the stocking quantity of shrimp is 900,000/ha. In 2012, we switched zones A and B, cultivating shrimp with grass carp in zone B but without fish in zone A. The stocking quantities of shrimp and fish were the same as in 2011. If a WSS outbreak occurred, shrimps were harvested immediately; if not, shrimps were harvested after 110 days of cultivation, and yields were measured.

**Validation of coculturing shrimp and African sharptooth catfish for controlling WSS in *L. vanmamei* farming**. In 2011, the polyculture system of coculturing *L. vanmamei* and African sharptooth catfish was validated at a farm in Qinzhou, Guangxi Province, China (Farm 2). Ninety-five farm ponds (88.2 ha) were divided into zone A and zone B. Zone A consisted of 38 ponds with a total area of 21.2 ha, and zone B consisted of 57 ponds with a total area of 67.0 ha. The stocking quantity of shrimp in the ponds of zone A is 750,000/ha. Shrimp were cultured in the ponds for 10 days before releasing African sharptooth catfish with an average body weight of 0.5 kg. The stocking quantity of fish is 525–750/ha. Shrimp were cultured without fish in the ponds of zone B, and the stocking quantity of shrimp is 750,000/ha. In 2012, we split zone B into zones B1 and B2. Shrimp were cultivated with catfish in 38 ponds of zone A and 25 ponds (27.00 ha) of zone B1, while shrimp were cultivated without fish in 32 ponds (40.00 ha) of zone B2. The stocking quantities of shrimp and fish were the same as in 2011. If WSS outbreak occurred, shrimps were harvested immediately; if not, shrimps were harvested after 110 days of cultivation, and yields were measured.

**Long-term validation of coculturing shrimp and fish for controlling WSS in *L. vanmamei* cultivation**. We tested the effectiveness of using fish for controlling WSS in shrimp production at a farm in Maoming, Guangdong Province, China (Farm 1) from 2013 to 2019. In 2013, shrimp were co-cultured with African sharptooth catfish of body weight ranging from 0.5 to 0.6 kg in 13 ponds (3.73 ha). The stocking quantity of shrimp in these ponds ranges from 878,788/ha to 1,230,769/ha. And shrimp were co-cultured with grass carp of body weight ranges from 0.7 to 1.0 kg and African sharptooth catfish of body weight ranges from 0.5 kg to 0.6 kg in 10 ponds (3.7 ha). The stocking quantity of shrimp in these ponds ranges from 909,091/ha to 1,212,121/ha. Additionally, shrimp were cultured without fish in 11 ponds (3.63 ha). The stocking quantity of shrimp in these ponds ranges from 878,788/ha to 969,697/ha. If WSS outbreak occurred, shrimp were harvested immediately; if not, shrimp were harvested after 110 days of cultivation.

In 2014, shrimp were cocultured with grass carp of body weight ranging from 0.7 to 1.0 kg in 8 ponds (2.76 ha). The stocking quantity of shrimp in these ponds ranges from 833,333/ha to 1,060,606/ha. And shrimp were co-cultured with grass carp of body weight ranges from 0.7 to 1.0 kg and African sharptooth catfish of body weight ranges from 0.5 to 0.6 kg in 12 ponds (4.03 ha). The stocking quantity of shrimp in these ponds ranges from 825,000/ha to 1,060,606/ha. Additionally, shrimp were cultured without fish in 5 ponds. The stocking quantity of shrimp in these ponds was 1,060,606/ha. If a WSS outbreak occurred, shrimp were harvested immediately; if not, shrimp were harvested after 110 days of cultivation.

In 2015, shrimp were co-cultured with grass carp of body weight ranging from 0.7 to 1.0 kg in 19 ponds (7.4 ha). The stocking quantity of shrimp in these ponds ranges from 746,269 to 1,538,462/ha. In addition, shrimp were cultured without fish in 10 ponds (3.8 ha). The stocking quantity of shrimp in these ponds ranges from 750,000/ha to 909,091/ha. If a WSS outbreak occurred, shrimp were harvested immediately; if not, shrimp were harvested after 110 days of cultivation.

In 2016, shrimp were co-cultured with grass carp of body weight ranging from 0.7 to 1.0 kg in 19 ponds (8.11 ha). The stocking quantity of shrimp in these ponds ranges from 488,372/ha to 636,364/ha. Additionally, shrimp were cultured without fish in 8 ponds (2.84 ha). The stocking quantity of shrimp in these ponds ranges from 543,478/ha to 636,364/ha. If a WSS outbreak occurred, shrimp were harvested immediately; if not, shrimp were harvested after 110 days of cultivation.

In 2017, shrimp were cocultured with grass carp of body weight ranging from 0.7 to 1.0 kg in 6 ponds (1.56 ha). The stocking quantity of shrimp in these ponds was 961,538/ha. And shrimps were co-cultured with grass carp of body weight ranging from 0.7 kg to 1.0 kg and African sharptooth catfish of body weight ranges from 0.5 to 0.6 kg in 12 ponds (3.96 ha). The stocking quantity of shrimp in these ponds ranges from 848,485/ha to 909,091/ha. Additionally, shrimp were cultured without fish in 9 ponds (2.76 ha). The stocking quantity of shrimp in these ponds ranges from 848,485/ha to 961,538/ha. If a WSS outbreak occurred, shrimp were harvested immediately; if not, shrimp were harvested after 110 days of cultivation.

In 2018, shrimp were cocultured with grass carp of body weight ranging from 0.7g to 1.0 kg in 22 ponds (9.24 ha). The stocking quantity of shrimp in these ponds ranges from 454,545/ha to 869,565/ha. Additionally, shrimp were cultured without fish in 9 ponds (3.36 ha). The stocking quantity of shrimp in these ponds ranges from 695,652/ha to 861,111/ha. If a WSS outbreak occurred, shrimp were harvested; if not, shrimp were harvested after 110 days of cultivation.

In 2019, shrimp were cocultured with grass carp of body weight ranging from 0.7 to 1.0 kg in 30 ponds (11.31 ha). The stocking quantity of shrimp in these ponds ranges from 652,174/ha to 1,000,000/ha. Additionally, shrimp were cultured without fish in 10 ponds (3.57 ha). The stocking quantity of shrimp in these ponds ranges from 666,667/ha to 1,000,000/ha. If a WSS outbreak occurred, shrimp were harvested immediately; if not, shrimp were harvested after 110 days of cultivation.

**Validation of coculturing shrimp and brown-marbled grouper for controlling WSS in *P. monodon* farming**. In 2013, the polyculture system of coculturing *P. monodon* and brown-marbled grouper was validated at a farm in Changjiang, Hainan Province, China (Farm 3). We cultured $6 \times 10^5$/ha of non-SPF shrimp in 6 ponds (1.60 ha) for 30 days and then introduced 600~750/ha of brown-marbled grouper with an average body weight of 0.1 kg. Shrimp were also cultured without fish in 3 ponds (0.8 ha). The stocking quantity of shrimp in these ponds is $6 \times 10^5$/ha. If a WSS outbreak occurred, shrimp were harvested immediately; if not, shrimp were harvested after 150 days of cultivation, and yields were measured.

In 2014, we cultured $6 \times 10^5$/ha of non-SPF shrimp in 6 ponds (1.60 ha) for 30 days and then introduced 600–750/ha of brown-marbled grouper with an average body weight of 0.1 kg. Shrimps were also cultured without fish in 3 ponds (0.8 ha). The stocking quantity of shrimp in these ponds is $6 \times 10^5$/ha. If a WSS outbreak occurred, shrimp were harvested immediately; if not, shrimp were harvested after 150 days of cultivation, and yields were measured.

**Validation of coculturing shrimp and branded gobies for controlling WSS in *M. japonica* farming**. In 2013, the polyculture system of coculturing *M. japonica* and branded gobies was validated at a farm in Qingdao, Shandong Province, China (Farm 4). We cultured $1.5 \times 10^5$/ha of non-SPF shrimp in 10 ponds (13.40 ha) for 30 days and then introduced 750~900/ha of branded gobies with an average body weight of 0.05 kg. Shrimp were also cultured without fish in 5 ponds (6.70 ha). The stocking quantity of shrimp in these ponds is $1.5 \times 10^5$/ha. If a WSS outbreak occurred, shrimp were harvested immediately; if not, shrimp were harvested after 100 days of cultivation, and yields were measured.

In 2014, we cultured $1.5 \times 10^5$/ha of non-SPF shrimp in 10 ponds (13.40 ha) for 30 days and then introduced 750~900/ha of branded gobies with an average body weight of 0.1 kg. Shrimp were also cultured without fish in 5 ponds (6.70 ha). The stocking quantity of shrimp in these ponds is $1.5 \times 10^5$/ha. If a WSS outbreak occurred, shrimp were harvested immediately; if not, shrimp were harvested after 100 days of cultivation, and yields were measured.

**Promotion of the polyculture system at a farmers' association in Nansha, China**. When we promoted the polyculture system at the farmers' association in 2015, only 6 farmers decided to adopt the system, as most of the farmers worried

that fish would ingest healthy shrimp. Each of the 6 farmers introduced 225,000, 360,000, and 360,000 *P.monodon* postlarva to his/her earthen pond (3 ha) on March 28, May 8, and June 15, respectively. And 1350 grass carps with an average body weight of 1 kg were released in the ponds on April 30. These farmers harvested shrimp from May to November, and grass carp on December 14. The yields of shrimp and fish of these six ponds were recoded. The other farmers in the association introduced 225,000 and 360,000 of *P.monodon* postlarva to their ponds (3 ha) on March 28 and May 8, respectively. WSS outbreaks occur in their ponds from May 15 to May 23. Therefore, these farmers only harvested shrimp in May. Six ponds were randomly selected, and the yields of these ponds were recorded.

**Promotion of the polyculture system at a farmers' association in Tanghai, China**. Farmers at the farmers' association used to culture 1500/ha of *F. chinensis* in earthen pond (5 ha) before the promotion of the polyculture system in 2015. The yields of 10 randomly selected ponds in 2014 were recorded. In 2015, farmers at the association started to culture 8,000/ha of *F. chinensis* in their ponds. The shrimp were cultured 20 days before 800/ha of branded gobies with an average body weight of 0.05 kg were released in the ponds. Branded gobies were cultivated for 15 days before introducing to the ponds. Shrimps were harvested after 120 days of cultivation. The yields of ten randomly selected ponds were recorded.

**Statistics and reproducibility**. Alpha levels of 0.05 were regarded as statistically significant throughout the study. Three replicates were set up for each experiment to confirm the reproducibility of the data. All data are reported as the mean ± standard errors.

**Reporting summary**. Further information on research design is available in the Nature Research Reporting Summary linked to this article.

## Data availability
All data is available in the main text or the Supplementary Materials.

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

## Acknowledgements
This research was supported by the Chinese Agriculture Research System (No. CARS-47), and the National Natural Science Foundation of China (No. U1131002), the National Key Technology R&D Program (2012BAD17B03), the Special Fund for Agro-scientific Research in the Public Interest (No. 201103034), and the National Basic Research Program of China (No. 2012CB114401). The funders had no role in study design, data collection and analysis, decision to publish, or preparation of the paper.

## Author contributions

J.G.H. conceived of the project and designed research, S.C.L. and G.C.F. helped the proceeding. Y.G.C. and S.P.W. were the lead coordinators for laboratory and field study; Y.G.C., J.C.Y., S.Y.L., C.L., F.H.Y., Z.W.Z., H.Q.Z., M.Y.Z., L.R.L., K.Y., C.L.G., X.J.L., Y.J.L. and R.J.L. performed the experiments; Z.L.Y., B.B.M. and L.W.Z. performed the field study; M.W. and B.A. analyzed the data; Z.Z. and G.C.F. performed the mathematical modeling; M.W., S.C.L., and J.G.H. wrote the paper with the contribution from all authors.

## Competing interests

The authors declare no competing interests.
