## [Transparent Peer Review File · Communications Biology]

Reviewers' comments:

Reviewer #1 (Remarks to the Author):

At first impression, the manuscript appears to be one more polyculture study. However, after a detailed review, it can be seen that it is a robust study in which different scenarios were considered to obtain the best possible conditions regarding the ideal fish species to avoid the spread of the virus. Subsequently, the results obtained from the investigation were tested on a larger scale, in which the theory that fish in polyculture decrease the transmission of WSSV was confirmed.

One of the features of the proposed system is to maintain low salinity levels according to the fish requirements, which is very different from that detected in most shrimp farms around the world (35-40 PSU); in this regard, the implementation of this system may not be feasible for most users. I consider it necessary to discuss the lack of polyculture studies using marine species or those that tolerate higher salinity. In this study, species that tolerate low salinities were used; However, most of the open farms that produce penaeid shrimp show salinities above those registered in the marine environment. Perhaps propose studies with estuarine species, etc. I also suggest including this reference: Shrimp polyculture: a potentially profitable, sustainable, but uncommon aquacultural practice. Reviews in Aquaculture. 2(2):73-85, which is one of the first reports aiming to concatenate and analyze the available information about shrimp polyculture. The potential risks of introducing fish into the ponds should be discussed. Which measures should be considered before introducing any species into a shrimp farm?. I think that fish should be certified by some agency. It seems that the document fails to discuss and propose which of all the proven systems turns out to be the most recommended and why. The discussion mostly addresses this practice's social and ecological benefits, but very little addresses the results obtained. Higher size letters have to be used in figures. These are difficult to read. Finally, I'm unsure if the journal allows as many as four co-first and three co-corresponding authors. According to the author contributions section, J.G.H. should be the corresponding author.

Reviewer #2 (Remarks to the Author):

Wang et al. developed a polyculture system which that controlled the outbreaks of white spot syndrome (WSS), which is a severe viral disease in shrimp aquaculture. Based on experiments and mathematical modelling, they developed a convenient cultural system for small-scale farmers, who cultivate shrimps in earthen ponds without biosecurity measures. Additionally, the paper demonstrated selective predation could control a disease with high transmission rate, which is important for applied Ecology. This work is comprehensive and should bring interests to the readers of Communications Biology. However, I have a few minor comments:

1. In line 152, the authors should state more clearly whether Model 3 is suitable for all fish species? What do other researchers need to do if they want to develop their own system using Model 3?
2. In line 169, the authors cultured shrimps in 18 ponds. They did not observe WSS outbreaks in only 17 ponds. Did WSS occur in the other pond?
3. In line 205, branded goby is not an aquaculture species. Do the farmers need to apply some special treatments for using this species in the system?
4. In line 235, how did the authors determine the stocking density of *F. chinensis* before their promotion of the polyculture system.
5. The time window for controlling WSS is not clearly indicated in Supplementary Figure 4.

Reviews' comments:

Reviewer #1 (Remarks to the Author)

At first impression, the manuscript appears to be one more polyculture study. However, after a detailed review, it can be seen that it is a robust study in which different scenarios were considered to obtain the best possible conditions regarding the ideal fish species to avoid the spread of the virus.

Subsequently, the results obtained from the investigation were tested on a larger scale, in which the theory that fish in polyculture decrease the transmission of WSSV was confirmed.

1. One of the features of the proposed system is to maintain low salinity levels according to the fish requirements, which is very different from that detected in most shrimp farms around the world (35-40 PSU); in this regard, the implementation of this system may not be feasible for most users.

I consider it necessary to discuss the lack of polyculture studies using marine species or those that tolerate higher salinity. In this study, species that tolerate low salinities were used; However, most of the open farms that produce penaeid shrimp show salinities above those registered in the marine environment. Perhaps propose studies with estuarine species, etc.

Response: Thank you for your excellent advice. In the study, we determined the transmission dynamics of WSSV as well as the dynamics of the WSSV-infected shrimp through experiments and mathematical modelling (Model 1-3). Model 2 showed that there is a short time window (2 days) for preventing WSS outbreaks. As the time window is too short, introducing fish after observing dead WSSV-infected shrimp in the ponds cannot control the spread of WSSV. Thus, it is necessary to co-culturing

shrimp with fish, which allows the fish to remove the moribund shrimp promptly. Additionally, Model 3 demonstrated that polyculture system can effectively control WSS outbreaks in the cultivation of diverse cultivated shrimp if the fish can swallow the WSS-infected and dead shrimp promptly and completely. Thus, new systems can be developed based on the results of experiments and our mathematical models.

More than 50% shrimps were cultivated in the estuary areas of China. Farmers usually cultivate shrimps at low salinity in these areas. Therefore, we developed polyculture systems using grass carps and African sharptooth catfish, which are freshwater fish species. We also developed polyculture systems using two marine fish species (brown-marbled grouper and branded goby). Moreover, farmers and researchers can modify our system to adapt their own cultivation conditions. And we recommend the researchers to develop new polyculture systems using local aquaculture or native fish species. We discussed this in the revised manuscript from Line 300 to Line 323.

2. I also suggest including this reference: Shrimp polyculture: a potentially profitable, sustainable, but uncommon aquacultural practice. *Reviews in Aquaculture*. 2(2):73-85, which is one of the first reports aiming to concatenate and analyze the available information about shrimp polyculture.

Response: Thank you for your suggestion. We have cited the suggested reference and added one sentences in the revised manuscript at Line 72:

“Therefore, polyculture has been considered as a promising strategy in future sustainable shrimp aquaculture industry”.

3. The potential risks of introducing fish into the ponds should be discussed. Which measures should be considered before introducing any species into a shrimp farm? I think that fish should be certified by some agency.

Response: Thank you for your comments. We think there are two potential risks of introducing fishes in the ponds cultivating shrimps.

1) Exotic species

The fishes used in our polyculture systems are local aquaculture or native species in China. We suggested farmers or researchers to modify our system by using local aquaculture or native fish species according to their cultivation conditions. We agree that if the exotic fish species are used in the system. They should be certified by the local administration agency. We have discussed this in the revised manuscript from Line 317 to Line 320:

“Local aquaculture or native fish species are recommended to be used in the polyculture systems, as they are well-adapted to the local environment. The use exotic fish species in the polyculture system should be highly cautious and certified by local administration agency, as it might cause adverse environmental impacts.”

2) Diseases

It is reported that shrimps can be infected by a few fish diseases. Thus, the co-culture fish species should not carry pathogens that can infect shrimps. Otherwise, the fishes should be screened for these pathogens before introducing into the ponds. We have discussed this in the revised manuscript from Line 320 to Line 323:

“It is reported that shrimps can be infected by a few fish diseases. Thus, the co-culture fish species should not carry pathogens that can infect shrimps. Otherwise, the fishes should be screened for these pathogens before introducing into the ponds.”

4 It seems that the document fails to discuss and propose which of all the proven systems turns out to be the most recommended and why. The discussion mostly addresses this practice's social and ecological benefits, but very little addresses the results obtained.

Response: Thank you for your advice. We added two paragraphs to discuss the result in the revised manuscript from Line 300 to Line 323.

We described five polyculture system in the manuscript (see Supplementary

Note). The five systems are suitable for diverse species of shrimps that are cultivated under different conditions. The farmers and researchers can select one system according to their cultivation conditions. Additionally, researchers can develop new systems by changing the co-cultured fish to local aquaculture or native species, if the ones can swallow the WSSV-infected and dead shrimp promptly and completely. Therefore, we do not have the most recommended system. The users can select or develop a system that is suitable for their cultivation condition. We have discussed this in the revised manuscript from Line 311 to Line 315:

“Farmers usually cultivate shrimps at low salinity in the estuary areas of China, where produces more than 50% of shrimps in China. Therefore, two freshwater fish species, grass carp and catfish, were used in our polyculture systems. We also developed polyculture systems using brown-marbled grouper and branded goby, which tolerate high salinities. Farmers can select a polyculture system that is suitable for their cultivation conditions.”

5. Higher size letters have to be used in figures. These are difficult to read.

Response: Thank you for your suggestions. We have increased the font size in the figures.

6. Finally, I'm unsure if the journal allows as many as four co-first and three co-corresponding authors. According to the author contributions section, J.G.H. should be the corresponding author.

Response: Thank you for your comments. To develop the polyculture system, we combined the studies of epidemiology and ecology. In addition, we derived three mathematical models to better investigate the transmission dynamics of WSSV and the dynamic of WSSV-infected shrimp. For the co-corresponding authors, S.C.L. supervised the ecological analyses, G.C.F. supervised the development of mathematical models, and J.G.H. supervised the study of epidemiology. The three co-corresponding

authors had lots of discussions during ten years of the study.

For the co-first authors, M.W. analyzed the data and wrote the manuscript, Y.G.C. was the lead person of the experiments, Z.Z. derived the mathematical models, S.P.W. was the lead person for promoting the polyculture system.

Thus, we believe that it is necessary to have four co-first and three co-corresponding authors.

Reviewer #2 (Remarks to the Author)

Wang et al. developed a polyculture system which that controlled the outbreaks of white spot syndrome (WSS), which is a severe viral disease in shrimp aquaculture. Based on experiments and mathematical modelling, they developed a convenient cultural system for small-scale farmers, who cultivate shrimps in earthen ponds without biosecurity measures. Additionally, the paper demonstrated selective predation could control a disease with high transmission rate, which is important for applied Ecology. This work is comprehensive and should bring interests to the readers of Communications Biology. However, I have a few minor comments:

1. In line 152, the authors should state more clearly whether Model 3 is suitable for all fish species? What do other researchers need to do if they want to develop their own system using Model 3?

Response: Thank you for your advice. Model 3 is suitable for diverse species of fishes. We have added one sentence in the revised manuscript at Line 159:

“In addition, the capacity of fish for preventing WSS outbreaks can be determined by Model 3 (**Supplementary Note**), which is suitable for diverse species of fishes”

Two parameters are needed if the researchers want to develop their own system using Model 3: 1) fish-feeding quantity of dead shrimp, 2) fish-feeding ratio of ratio of dead shrimp over healthy shrimp. We described these two parameters at Line 88 of the supplementary materials.

2. In line 169, the authors cultured shrimps in 18 ponds. They did not observe WSS outbreaks in only 17 ponds. Did WSS occur in the other pond?

Response: Thank you for your comment. The pond is unsuccessful due to pathogenic bacterium (*Vibrio*) infection. We confirmed that WSS did not occur in the pond through a one-step PCR assay. We have added one sentence in the revised manuscript at Line

174:

“One pond was unsuccessful due to pathogenic bacterium (*Vibrio*) infection.”

3. In line 205, branded goby is not an aquaculture species. Do the farmers need to apply some special treatments for using this species in the system?

Response: Thank you for your comment. Branded goby is a native species in Bohai Sea. The ponds using branded gobies as the co-cultured species are at the coast of Bohai Sea. Thus, branded gobies can be easily found in the area. We recommend the farmers to culture branded gobies for 15 days before introducing into the ponds. We have added one sentence in the method Section of the revised manuscript at Line 652:

“Branded gobies were cultivated for 15 days before introducing to the ponds.”

4. In line 235, how did the authors determine the stocking density of *F. chinensis* before their promotion of the polyculture system.

Response: Thank you for your comment. We determined the stocking density according to the production log of the farmers' association.

5. The time window for controlling WSS is not clearly indicated in Supplementary Figure 4.

Response: Thank you for your suggestion. We have modified the figure to highlight the time window (see Figure in the next page).

REVIEWERS' COMMENTS:

Reviewer #1 (Remarks to the Author):

I consider that the responses to my observations were addressed and resolved in an appropriate way. Therefore, I have no additional observations. I recommend acceptance of the manuscript.

Reviews' comments:

Reviewer #1 (Remarks to the Author)

I consider that the responses to my observations were addressed and resolved in an appropriate way. Therefore, I have no additional observations.

I recommend acceptance of the manuscript.

Response: Thank you so much!